# Longitudinal fundus imaging and its genome-wide association analysis provide evidence for a human retinal aging clock

**Sara Ahadi**[1]*, **Kenneth A Wilson**[2‡], **Boris Babenko**[3‡], **Cory Y McLean**[4]*‡, **Drew Bryant**[1], **Orion Pritchard**[1], **Ajay Kumar**[5], **Enrique M Carrera**[2], **Ricardo Lamy**[6], **Jay M Stewart**[7], **Avinash Varadarajan**[3], **Marc Berndl**[1], **Pankaj Kapahi**[2]*†, **Ali Bashir**[1]†

[1]Google Research, Mountain View, United States; [2]Buck Institute for Research on Aging, Novato, United States; [3]Google Health, Palo Alto, United States; [4]Google Health, Cambridge, United States; [5]Department of Biophysics, Post Graduate Institute of Medical Education and Research, Chandigarh, India; [6]Department of Ophthalmology, Zuckerberg San Francisco General Hospital and Trauma Center, San Francisco, United States; [7]Department of Ophthalmology, University of California, San Francisco, San Francisco, United States

***For correspondence:**
saraahadi@gmail.com (SA);
cym@google.com (CYMcL);
Pkapahi@buckinstitute.org (PK)

†These authors contributed
equally to this work
‡These authors also contributed
equally to this work

**Abstract** Biological age, distinct from an individual's chronological age, has been studied extensively through predictive aging clocks. However, these clocks have limited accuracy in short time-scales. Here we trained deep learning models on fundus images from the EyePACS dataset to predict individuals' chronological age. Our retinal aging clocking, 'eyeAge', predicted chronological age more accurately than other aging clocks (mean absolute error of 2.86 and 3.30 years on quality-filtered data from EyePACS and UK Biobank, respectively). Additionally, eyeAge was independent of blood marker-based measures of biological age, maintaining an all-cause mortality hazard ratio of 1.026 even when adjusted for phenotypic age. The individual-specific nature of eyeAge was reinforced via multiple GWAS hits in the UK Biobank cohort. The top GWAS locus was further validated via knockdown of the fly homolog, *Alk*, which slowed age-related decline in vision in flies. This study demonstrates the potential utility of a retinal aging clock for studying aging and age-related diseases and quantitatively measuring aging on very short time-scales, opening avenues for quick and actionable evaluation of gero-protective therapeutics.

## Editor's evaluation

This paper is an important contribution to the biological aging field using eye image data to create an aging clock of the retina in data from eyePACS with validation in the UK Biobank. The authors provide compelling evidence that the clock correlates with chronological and phenotypic age, predicting mortality independently of chronological age and showing longitudinal evidence. The work identifies novel genetic loci with a top site located in the ALKAL2 region, which is functionally validated in a *Drosophila* model.

## Introduction

Aging causes molecular and physiological changes throughout all tissues of the body, enhancing the risk of several diseases (*López-Otín et al., 2013*). Identifying specific markers of aging is a critical area of research, as each individual ages uniquely depending on both genetic and environmental factors (*Ahadi et al., 2020*). While a variety of aging clocks have recently been developed to track the aging

process, including phenotypic age (*Liu et al., 2018*; a combination of chronological age and nine biomarkers predictive of mortality) and epigenetic clocks derived from DNA methylation (*Horvath and Raj, 2018*), many require a blood draw and multiplex assay of many analytes.

A growing body of evidence suggests that the microvasculature in the retina might be a reliable indicator of the overall health of the body's circulatory system and the brain. Changes in the eyes accompany aging and many age-related diseases such as age-related macular degeneration (AMD) (*Luu and Palczewski, 2018*), diabetic retinopathy (*Namperumalsamy et al., 2009*), and neurodegenerative disorders like Parkinson's (*Luu and Palczewski, 2018*; *Archibald et al., 2009*) and Alzheimer's (*Frost et al., 2013*). Eyes are also ideal windows for early detection of systemic diseases by ophthalmologists, including AIDS (*Sun et al., 2009*; *Cunningham and Margolis, 1998*), chronic hypertension (*Wong and McIntosh, 2005*), and tumors (*Kreusel et al., 2002*). This broad utility is perhaps unsurprising, as any subtle changes in the vascular system first appear in the smallest blood vessels, and retinal capillaries are amongst the smallest in the body.

The subtle changes induced in these small vessels often go undetected by even the most sophisticated instruments, necessitating the use of better approaches involving deep learning. Fundus imaging has proven to be a powerful and non-invasive means for identifying specific markers of eye-related health. Deep-learning was initially employed to predict diabetic retinopathy from retinal images at accuracies matching, or even exceeding, experts (*Gulshan et al., 2016*). Since then, retinal images have been employed to identify at least 39 fundus diseases including glaucoma, diabetic retinopathy, age-related macular degeneration (*Wong and McIntosh, 2005*; *Cen et al., 2021*), cardiovascular risk (*Poplin et al., 2018*), chronic kidney disease (*Sabanayagam et al., 2020*), and, most recently, in predicting age (*Zhu et al., 2023*). Given its non-invasive, low-cost nature, retinal imaging provides an intriguing opportunity for longitudinal patient analysis to assess the rate of aging.

Here, we use deep learning models to predict chronological age from fundus retinal images, hereafter 'eyeAge', and use the deviation of this value from chronological age, hereafter 'eyeAgeAccel',

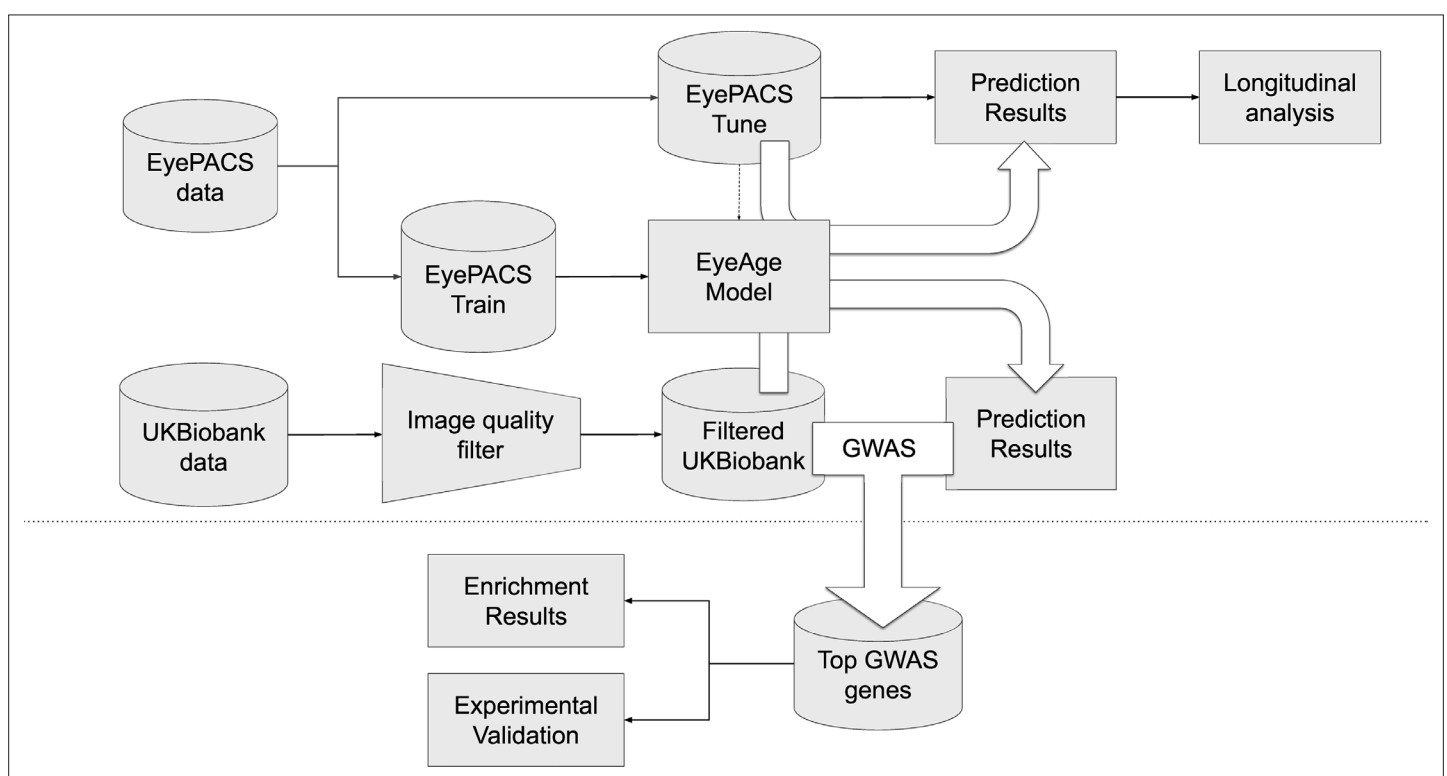

**Figure 1.** Schematic of analysis pipeline. EyePACS images were split into train and tune sets based on the patient. The model was then trained with the final model step being selected via the tune set. Prediction results on the EyePACS tune set were used for longitudinal analysis of aging. After filtering for image quality, inference was performed with the same model on the UK Biobank dataset and filtering for image quality, and the resulting eyeAgeAccel was used for GWAS analysis. Enrichment analysis was performed on the GWAS hits with a homolog of the top gene (*ALKAL2*) validated experimentally in *Drosophila*.

**Table 1.** Characteristics of patients in the development and validation sets (before filtering).

| | Development set (EyePACS) | | Test set (UK Biobank) |
|---|---|---|---|
| | **Train** | **Tune** | |
| Patients | 100,692 | 25,238 | 64,019 |
| Images | 217,289 | 54,292 | 119,532 |
| Ethnicity | Black: 11908 [7%]<br>Asia Pacific Islander: 11842 [7%]<br>White: 22539 [13%]<br>Hispanic: 125595 [71%]<br>Native American: 1791 [1%]<br>Other: 3809 [2%] | Black: 3040 [7%]<br>Asia Pacific Islander: 2923 [7%]<br>White: 5657 [13%]<br>Hispanic: 31521 [71%]<br>Native American: 426 [1%]<br>Other: 918 [2%] | Black: 1540 [1%]<br>Asia Pacific Islander: 4183 [4%]<br>White: 107967 [91%]<br>Hispanic: 0 [0%]<br>Native_american: 0 [0%]<br>Other: 5015 [4%] |
| Self-reported Sex | Female: 127075 [59%]<br>Male: 90128 [41%] | Female: 31743 [58%]<br>Male: 22531 [42%] | Female: 65739 [55%]<br>Male: 53793 [45%] |
| Age | median = 55.13<br>mean = 54.21<br>std = 11.50 | median = 55.19<br>mean = 54.20<br>std = 11.46 | median = 57.94<br>mean = 56.85<br>std = 8.18 |

for mortality and association analyses. We train this model on the well-studied EyePACS dataset and apply it on both the EyePACS and UK Biobank cohorts. Together, our results suggest that the trajectory of an individual's biological age can be predicted in timelines under a year and that statistically significant genome-wide associations are possible. Enrichment analysis of top GWAS hits as well as experimental validation of the *Drosophila* homolog of *ALKAL2*, a gene in the top GWAS locus, indicates genetic markers of visual decline with age and demonstrates the potential predictive power of a retinal aging clock in assessing biological age.

## Results
### Prediction of age from fundus images

*Figure 1* summarizes the analysis workflow for the study. Using the EyePACS dataset, we trained a fundus image model on 217,289 examples from 100,692 patients and tuned it on 54,292 images from 25,238 patients. These models were employed for longitudinal analysis of repeat patients and also applied on the UK Biobank dataset (119,532 images) which had a notably distinct demographic distribution (*Table 1*). For both studies, most visits generated two images, one image each for the left and right eye, the EyePACs dataset had more repeat visits by patients making the ratio of total images to total patients slightly larger (*Table 1*). In both analyses, we took the average of the predictions between the left and right eye from a single visit to infer age (See Materials and methods).

The model showed a strong correlation between chronological age and predicted age (eyeAge) in both the EyePACS (0.95) and UK Biobank (0.87) datasets (*Figure 2—figure supplement 1*). Using mean absolute error (MAE) to assess the fidelity of the aging clock showed that the model performed favorably on both datasets (2.86 and 3.30, respectively, after quality filtering) relative to previous studies (*Zhu et al., 2023*; *Galkin et al., 2021*; *McEwen et al., 2020*; *Horvath, 2013*). Next, we evaluated the efficacy of our predictions in one to two year time scales using longitudinal data. Using the EyePACS Tune dataset, we restricted ourselves to data from patients with exactly two visits (1719 subjects) and examined the models' ability to order the two visits over multiple time scales. Note that no longitudinal information about patients was specifically used to train or tune the model to predict chronological age. While the observed and predicted age differences between the two visits (M=0.033, SD = 2.34, *Figure 2—figure supplement 2*) had low correlation (pearson ρ=0.17, p-value = 1.4e-12), *Figure 2A* shows that the model correctly ordered 71% of visits within a year with an MAE less than 2 years. In both metrics, the fidelity decreased in older groups and with smaller age gaps.

To understand if this effect was simply a result of the noise of our innate age prediction, we performed an age-matched control experiment. We compared correlations between data points of one individual to data from a random pair of age-matched individuals (see Materials and methods).

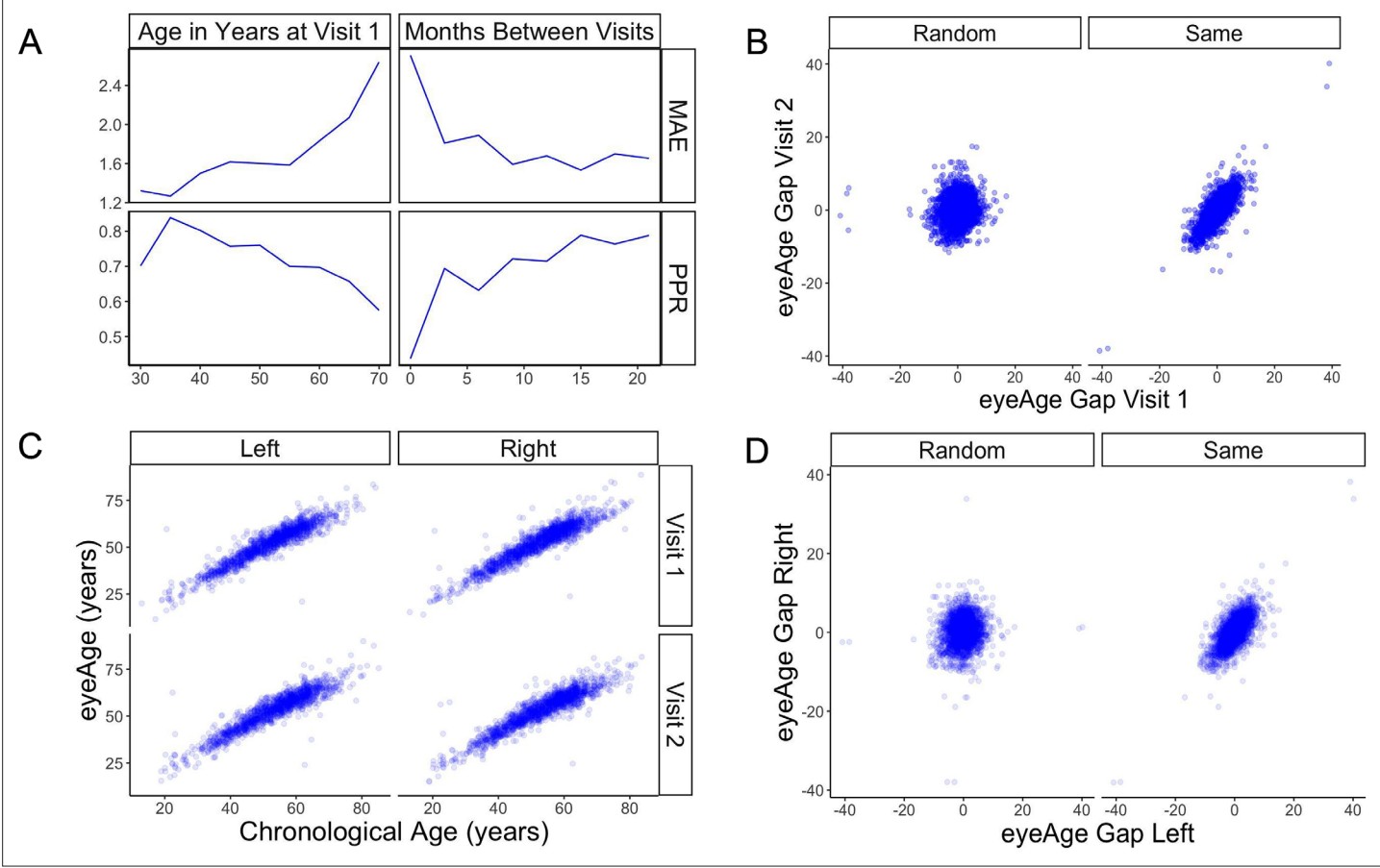

**Figure 2.** Longitudinal analysis of patients with exactly two visits in the EyePACS cohort. (**A**) Changes of PPR (positive prediction ratio: the ratio of data whose eyeAge increased between subsequent visits) and MAE (mean absolute error) calculated on the same individual in relationship to chronological age at the first visit (left) and time between longitudinal visits (right). (**B**) Scatter plots representing correlation between eyeAge Gap (difference between predicted age and chronological age) of two consecutive visits from an individual (Same) or two consecutive visits from two different individuals (Random). (**C**) Correlation of eyeAge and chronological age between left and right and two consecutive visits of the same individual. (**D**) Scatter plots representing the correlation of left and right eyeAge Gap from the same or two random individuals.

The online version of this article includes the following source data and figure supplement(s) for figure 2:

**Source data 1.** MAE and positive prediction ratio for time-matched and random individuals based on age at visit 1.

**Source data 2.** MAE and positive prediction ratio for time-matched and random individuals based on months between visits.

**Source data 3.** Age gap for random and time-matched individuals at visit 1 and 2.

**Source data 4.** Chronological and predicted age for left and right eye.

**Source data 5.** Age gap for random and time-matched individuals for left and right eyes.

**Source data 6.** Scatter plot of eyeAge with chronological age.

**Figure supplement 1.** Scatter plot of eyeAge with chronological age (Pearson ρ=0.96).

**Figure supplement 2.** Scatterplot showing the time elapsed (x-axis) vs. the difference between time elapsed and change in eyeAge (y-axis).

**Figure supplement 3.** Positive prediction ratio and MAE for random, time-matched individuals.

Comparisons were performed between each eye and timepoint. For all comparisons, the robust correlation observed within an individual's data was lost in data between time-matched individuals (*Figure 2B and D*). Additionally, the positive predictive ratio and MAE exhibited reduced performance, 55% and 3.6 years (*Figure 2—figure supplement 3*), suggesting a reproducible, individual-specific eyeAge component. To further explore this individual-specific component, *Figure 2C* compares eyeAge and chronological age within an individual between eyes and timepoints, showing strong correlation in each quadrant.

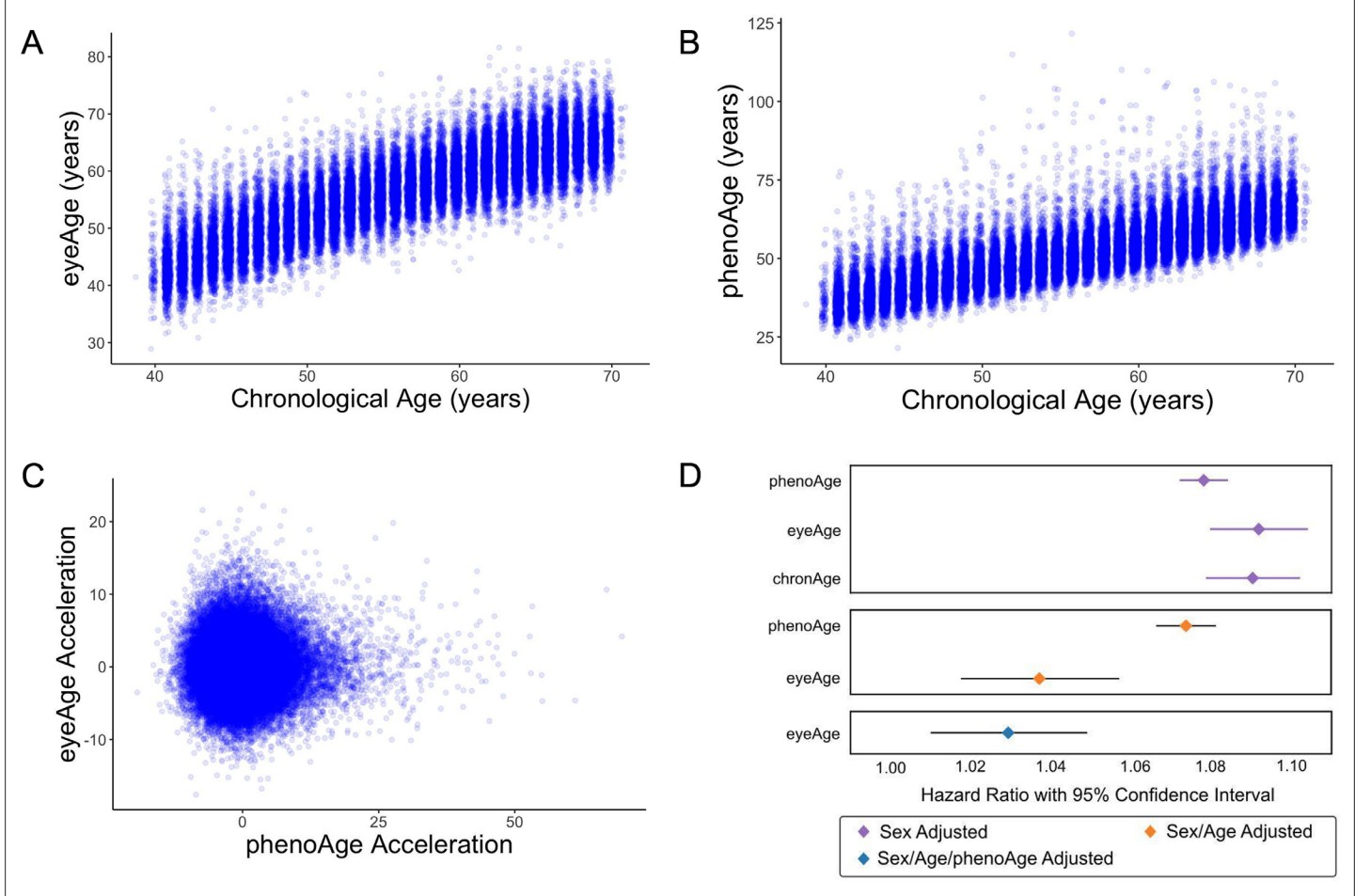

**Figure 3.** Relationships between eyeAge, phenoAge, and chronological age in the UK Biobank cohort. (**A**) Correlation between eyeAge and chronological age (Pearson ρ=0.86). (**B**) Correlation between phenoAge and chronological age (Pearson ρ=0.82). (**C**) Correlation between eyeAgeAcceleration and phenoAgeAcceleration (Pearson ρ=0.12). (**D**) Forest plot of all-cause mortality hazard ratios (diamonds) and confidence intervals (lines) for the UK Biobank dataset. Purple lines are adjusted only for sex; orange lines are adjusted for sex and age; blue lines are adjusted for sex, age, and phenoAge.

The online version of this article includes the following source data and figure supplement(s) for figure 3:

**Source data 1.** Age, eyeAge, phenoAge, eyeAge Acceleration and phenoAge Acceleration values for each individual.

**Figure supplement 1.** Scatter plot of eyeAge and phenoAge (Pearson ρ=0.71).

**Figure supplement 2.** eyeAge hazard ratio adjusted with and without visual acuity.

## Testing the model in UK Biobank cohort

We next applied our EyePACS-trained eyeAge model to the UK Biobank dataset. The UK Biobank cohort included retinal fundus images from 64,019 patients as well as extensive clinical labs and genomic data. These clinical markers enabled comparison of eyeAge with phenoAge, a clinical blood marker-based aging clock (*Liu et al., 2018*). The observed 0.87 correlation between eyeAge and chronological age in the UK Biobank cohort was consistent with (and slightly higher than) the observed correlation of phenoAge and chronological age (0.82) (*Figure 3A and B*). Notably, the correlation between phenoAge and eyeAge was substantially lower (0.72; *Figure 3—figure supplement 1*) and, in fact, roughly equivalent to the product of their respective correlations with chronological age, suggesting that they were largely independent. To explore this further, we computed the residuals from linear models that independently regressed chronological age on phenoAge and eyeAge, as described previously (*Liu et al., 2018*), yielding phenoAge acceleration (phenoAgeAccel) and eyeAge acceleration (eyeAgeAccel), and observed little correlation between the two age acceleration measures (*Figure 3C*). We then performed Cox proportional hazards regression analysis to assess

mortality risk (*Cox, 1972*). The hazard ratio for eyeAge was statistically significant when adjusting for (self-reported) sex (1.09, CI=[1.08, 1.10], p-value = 1.6e-53), sex and age (1.04, CI=[1.02, 1.06], p-value = 1.8e-4), and sex, age, and phenoAge (1.03, CI=[1.01, 1.05], p-value = 2.8e-3) (*Figure 3D*). Stratifying the hazard ratio analysis showed a slight increase in the hazard ratio for women compared to men (1.035 vs 1.026), however the confidence intervals overlapped heavily (*Supplementary file 1*). Hazard ratio results adjusted for visual acuity are presented in (*Figure 3—figure supplement 2* and *Supplementary file 2*).

We also investigated the relationship between eyeAge and multiple additional measures of morbidity and disability available in the UK Biobank. We performed Cox proportional hazards regression on six additional chronic disease outcomes when adjusting for age and sex: chronic obstructive pulmonary disease (COPD), myocardial infarction, asthma, stroke, Parkinsonism, and dementia. Nominally significant associations between eyeAge and both COPD (p-value = 0.0048) and myocardial infarction (p-value = 0.049) were observed (*Supplementary file 3*). We performed linear regression on seven morbidity measurements reported at the time of imaging: fluid intelligence, systolic and diastolic blood pressure, the 'Health score (England)' index of multiple deprivation, pulse wave arterial stiffness, self-reported overall health rating, and self-reported presence of a longstanding illness. Increased eyeAgeAccel corresponded to significantly increased systolic blood pressure (p-value = 1.025e-7) and decreased levels of deprivation (p-value = 2.26e-5) as measured by the Health score (England) index of multiple deprivation (*Supplementary file 4*). Interestingly, increased eyeAgeAccel also corresponded with significantly increased performance in fluid intelligence scores (p-value = 5.34e-27).

As visual acuity has long been known to degrade with age (*Gittings and Fozard, 1986*), we examined the extent to which eyeAge explains the known correlation between chronological age and visual acuity. Although chronological age and eyeAge are highly correlated (*Figure 3A*), we observed a slightly higher correlation of eyeAge with visual acuity ($\rho$=0.221) compared to chronological age vs. visual acuity ($\rho$=0.218). Both measures of age appear relevant for visual acuity decline, as the influence of chronological age remained significant even after regressing out the influence of eyeAge on visual acuity (p-value = 1.6e-13, *Supplementary file 5*).

## GWAS and experimental validation of ALK

Based on the patient-specific eyeAgeAccel effects and its independence from phenoAgeAccel, a GWAS was conducted to identify genetic factors associated with eyeAgeAccel. We subsetted the cohort to individuals of European ancestry, performed genotype quality control, and utilized a single eyeAgeAccel value per individual, resulting in a cohort of 45,444 individuals for GWAS analysis. GWAS was performed using BOLT-LMM (see Materials and methods) with chronological age, sex, genotyping array type, the top five principal components of genetic ancestry, and indicator variables for the six assessment centers used for the imaging as covariates. Full GWAS summary statistics are available in *Supplementary file 6*.

Genomic inflation was low (1.05; *Figure 4—figure supplement 1*). The stratified linkage disequilibrium (LD) score regression-based intercept was 1.02 (SEM = 0.01), indicating that polygenicity, rather than population structure, drove the test statistic inflation. The SNP-based heritability was 0.11 (SEM = 0.02), an appreciable fraction of the estimated broad-sense heritability of biological age (27–57% via twin and family studies). The GWAS identified 38 independent suggestive hits ($R^2$ ≤0.1, p≤1 × $10^{-6}$) at 28 independent loci, 12 of which reached genome-wide significance (p≤5 × $10^{-8}$) (*Figure 4*, *Supplementary file 7*).

Many of the hits were associated with eye function and age-related disease (truncated list of candidate hits summarized in *Supplementary file 8*). The most significant locus spanned 650 kb and included three genes in a highly significant LD block: *SH3YL1*, *ACP1*, and *ALKAL2* (*Figure 4—figure supplement 2*). The *SH3YL1* gene has recently been implicated as a biomarker for nephropathy in type 2 diabetes (*Choi et al., 2021*), whereas *ALKAL2* enables protein tyrosine kinase activity (*Woodling et al., 2020*). In other significant gene candidates, we identified variants in the genes *OCA2*, *POC5*, and *GJA3*, which have all been implicated in eye development and function. *OCA2* specifically is known to be important for eye pigmentation (*Kamaraj and Purohit, 2014*), whereas *POC5* is linked to AMD (*Yan et al., 2018*). *GJA3* has been implicated in age-related cataract development (*Tang et al., 2019*). *MEF2C* has reported roles in numerous age-related conditions, including Alzheimer's

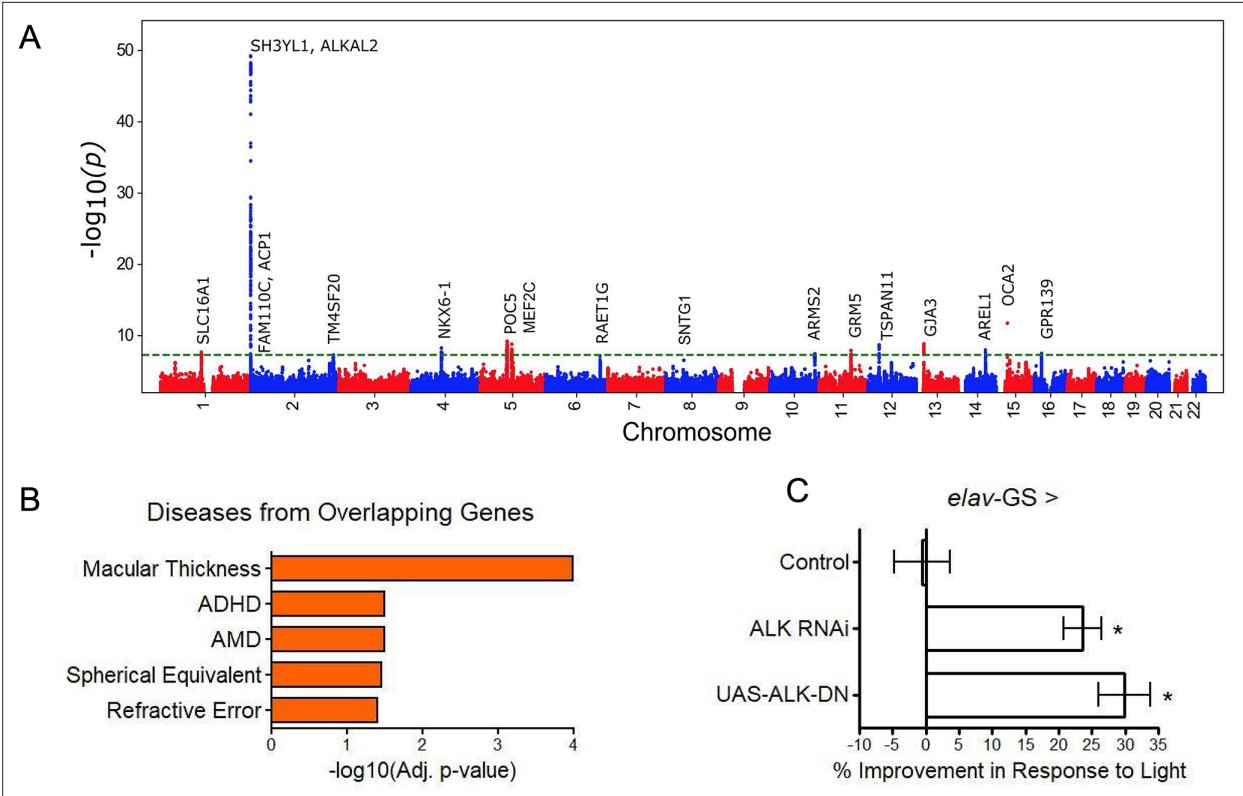

**Figure 4.** GWAS analyses and experimental validation. (**A**) Manhattan plot representing significant genes associated with eyeAgeAcceleration. (**B**) p-Values for enriched pathways: Macular thickness, ADHD (attention deficit hyperactivity disorder), AMD (age-related macular degeneration), spherical equivalent, and refractive error. (**C**) Assessment of visual performance of transgenic and control flies with age. p-Value is relative to control (*=p < 0.05). p-Value for ALK RNAi vs. control is 0.009; p-value for UAS-ALK-DN vs. control is 0.006. Error bars show standard deviation between 3 biological replicates. n = 100 flies per replicate.

The online version of this article includes the following figure supplement(s) for figure 4:

**Figure supplement 1.** eyeAgeAcceleration qq-plot.

**Figure supplement 2.** Zoom in on significant locus covering three genes in a highly significant LD block.

disease (*Xue et al., 2021*) and muscle wasting in cancer (*Judge et al., 2020*) and *GRM* is associated with age-related hearing loss (*Liu et al., 2021b*). Additional candidates are reported to be involved in cancer prognosis and progression, including *TSPAN11* (*Liu et al., 2021a*), *NKX6-1* (*Su et al., 2021*), and *SLC16A1* (*Zhang et al., 2021*).

Gene enrichment analysis (*Xie et al., 2021*) identified significant associations (adjusted p<0.05) between our gene candidates and macular thickness and degeneration, as seen in previous human GWAS studies (*Buniello et al., 2019*) and cataract formation (Elsevier pathway collection; *Cheadle et al., 2017*) as well as non-eye related diseases such as bone mineralization, tumor suppression, and Amyloid Precursor Protein pathways (Biocarta; *Nishimura, 2001*). Gene Ontology (GO) term analysis of our gene candidates revealed significant enrichment (adjusted p<0.05) for protein tyrosine kinase activator activity, gap junction channel activity, and wide pore channel activity (*Figure 4B*).

Sum of single effects regression (*Wang et al., 2020*) was used to identify putative causal variants for each locus (*Supplementary file 9*). In the most significant locus (*Figure 4—figure supplement 2*), we identified the deletion variant rs56350804 as the single variant with a posterior inclusion probability (PIP) above 0.45 (rs56350804 PIP = 0.9998). While rs56350804 is intronic to *SH3YL1*, expression quantitative trait locus (eQTL) analysis by the Genotype-Tissue Expression consortium identified significant eQTL between rs56350804 and each of *SH3YL1*, *ACP1*, and *ALKAL2* (GTEx Consortium 2020). In particular, the *ALKAL2* gene had its expression modulated by rs56350804 in cervical spinal cord tissue (p=3.0 × 10$^{-16}$), and inhibition of the *Drosophila* homolog of *ALKAL2*, *Alk*, has been shown

to extend lifespan (*Woodling et al., 2020*), making it a good candidate for exploring a potential role in visual function.

Previously, *D. melanogaster* has been used to study the impact of aging interventions on retinal health by using the phototaxis index, a fly's ability to be attracted toward light (*Hodge et al., 2022*). We used *D. melanogaster* to observe visual decline via phototaxis with transgenic *ALK* inhibition. We crossed the pan-neuronal RU486-inducible Gal4 driver *elav-Gal4*-GS with UAS-*Alk*^RNAi flies or UAS-*Alk*^DN to determine the effects of neuron-specific *Alk* inhibition. Both transgenic interventions resulted in significantly increased visual performance with age, whereas background controls showed no change in performance with RU486 treatment (*Figure 4C*). These results support the implication from the GWAS that *ALK* influences the aging of the visual system.

## Discussion

Retinal health has long been an important factor for visual aging, manifested as glaucoma, AMD, and other age-related retinal diseases, but until recently it was not known whether it could be indicative of overall health and aging. In this study, we applied deep learning models for predicting an individual's age from retinal fundus images and showed that these predictions may be informative for tracking aging patterns longitudinally. While other cellular and blood-related molecular markers of aging have recently been identified, these are at times invasive and, although accurate, take a long time to develop (*Horvath, 2013*). Other aging clocks from blood (*Horvath, 2013*; *Peters et al., 2015*), saliva (*Bocklandt et al., 2011*), skin (*Bocklandt et al., 2011*; *Fleischer et al., 2018*), muscle (*Mamoshina et al., 2018*), and liver (*Wang et al., 2017*) showed an MAE deviating 4–8 years from the actual age. More dynamic markers such as proteins and metabolites can track aging in shorter time intervals but are still limited to 2–4 years (*Ahadi et al., 2020*; *Wang et al., 2017*; *Chen et al., 2012*). In contrast, using deep learning models on retina fundus images, we were able to predict changes in aging at a granularity of less than a year. These small time-scales, and relative low-cost of imaging, makes eyeAge promising for longitudinal studies.

Correlation and hazard ratio analyses from our study suggest that eyeAge and phenotypic age are conditionally independent given chronological age. Therefore, eyeAge is a potential biomarker that reflects a layer of biological aging not included in blood markers. This is supported by our GWAS findings; different genes were associated with eyeAgeAccel compared to phenoAgeAccel (*Kuo et al., 2021*). However, there are limitations with this approach. Similar to other aging clocks (such as DNA-methylome), eyeAge underperforms phenotypic age in mortality prediction. This is likely because the biomarkers used to calculate phenotypic age were explicitly selected based on their ability to predict mortality. New algorithms that incorporate blood markers and retinal clocks have the potential to be better predictors of morbidity and mortality. Additionally, it remains to be seen whether eyeAgeAccel would reflect interventions such as behavioral changes or medication.

Our GWAS identified candidate genes associated with several eye- and age-related functions, such as *POC5* (*Yan et al., 2018*) and *GJA3* (*Tang et al., 2019*). Additional significant candidates had previously identified functions that are not restricted to the eye but are still related to age, e.g. *MEF2C* being associated with Alzheimer's disease (*Xue et al., 2021*) and multiple candidates (TSPAN11, NKX6-1, SLC16A1, RAET1G, SNTG1, ARRDC3, RASSF3, DIRC3, and GCNT3) associated with cancer (*Supplementary file 8*). These suggest that eyeAge may identify general signatures of aging rather than purely eye-related traits. Pathway analyses similarly were split between eye-related pathways and others that were not eye-specific. While we suspect many of the eye-related pathways to have an aging component, some pathways may be enriched artifactually. For example, though melanin biosynthesis has been associated with protection from photodamage (*Hodge et al., 2022*), the predicted quality of fundus images has also been shown to be influenced by eye color (*Guenther et al., 2020*). Notably, an independent group separately identified our top GWAS candidate locus as the most significant locus (*Goallec et al., 2021*). This combined with previous studies showing *ALK* to be important for lifespan extension in flies (*Woodling et al., 2020*) and our own experimental validation confirming improved ocular health in a fly homolog, *Alk,* is compelling evidence of a true biological signal in the GWAS.

Taken together, our work reinforces the utility of fundus imaging for evaluating overall health and opens up new opportunities for using it to predict longevity. eyeAge has substantial applications in aging and aging-related diseases, from biomarker application to tracking therapeutics. In particular, the retinal aging clock because of its ease of use, low cost, and non-invasive sample collection, has the unique potential to additionally assess lifestyle and environmental factors implicated in aging. Retinal aging clocks can be immensely valuable to future clinical trials of rejuvenation/anti-aging therapies and for personalized medicine to measure improvements in aging over short periods, not only improving actionability but also enabling rapid iteration.

# Materials and methods

## Key resources table

| Reagent type (species) or resource | Designation | Source or reference | Identifiers | Additional information |
|---|---|---|---|---|
| Strain, $w^{Dah}$ background (*Drosophila melanogaster*, females) | $w^{Dah}$ control strain | Laboratory of Linda Partridge, ***Woodling et al., 2020*** | | Maintained in Kapahi Lab |
| Strain, $w^{Dah}$ background (*Drosophila melanogaster*, females) | $UAS-ALK^{RNAi}$ RNAi for *ALK* | Laboratory of Linda Partridge, ***Woodling et al., 2020*** | VDRC GD 11446 | Maintained in Kapahi Lab |
| Strain, $w^{Dah}$ background (*Drosophila melanogaster*, females) | $UAS-ALK^{DN}$ dominant negative ALK overexpression | Laboratory of Linda Partridge, ***Woodling et al., 2020*** | | Maintained in Kapahi Lab |
| Strain, $w^{Dah}$ background (*Drosophila melanogaster*, females) | elav-GS Ru486 inducible Gal4 driver | Bloomington *Drosophila* Stock Center, ***Woodling et al., 2020*** | BDSC 43642 | Maintained in Kapahi Lab |
| Chemical compound, drug | RU486 (mifepristone) | United States Biological, ***Osterwalder et al., 2001*** | 282888 | For inducting fly GeneSwitch expression system; 200 µM final concentration in food |

## EyeAge model development

Model development was done on the EyePACS train dataset (***Table 1***). A deep learning model with an Inception-v3 architecture (***Deng et al., 2009***; ***Szegedy et al., 2015***) was trained to take a color fundus photo as input and predict the chronological age (referred to as chronologicalAge below) using L1 loss. Age values were normalized to have zero mean and unit variance before training (and during inference this normalization is reversed to get back to year units). Model training was stopped after 363,200 steps by looking at performance on the EyePACS tune dataset. The hyperparameters of the model were as follows: the initial learning rate was 0.0001, which was warmed up to 0.001 over 40,751 steps; after the warm up phase, the learning rate was decayed by a factor of 0.99 every 13,584 steps; dropout was applied to the prelogits at a rate of 0.2; a weight decay of 4e-5 was used. The model backbone was pre-trained using the ImageNet dataset (***Deng et al., 2009***). As some of the color fundus images in the UK Biobank dataset were of very low quality, we also trained a separate deep learning model to predict image quality, similar to what was reported in our prior work (***Mitani et al., 2020***; ***Varadarajan et al., 2018***).

## EyeAge model evaluation

The model described above was applied to images to predict chronological age. The image quality model described above was used to discard low quality images – reducing the initial 85,645 patient (174,049 image) dataset to 66,533 patients (120,362 images). Finally, we restricted the data to the first assessment visit to UK Biobank. This was done to reduce bias associated with image quality differences, as we observed quality differences between images captured in the later follow-up visits. Since these follow-up visits happened several years after the initial assessment, the time to event

or censorship is much smaller, and a model could exploit this association. For participants that had images of both eyes passing the quality filter, we averaged the predictions across the two eyes. After these processing steps, we ended up with 55,267 data points total, one per remaining participant. Next, using the predicted eyeAge and the chronologicalAge of the participant at the time of imaging, an 'eyeAgeAcceleration' score was calculated for each participant as the residuals of the ordinary least squares regression model 'chronologicalAge ~eyeAge' (*Liu et al., 2018*). In order to compare with another well-known biological marker of age, phenoAge (*Liu et al., 2018*) was also computed using the values of blood markers available for the participants. PhenoAgeAcceleration was then computed in an analogous manner to eyeAgeAcceleration.

## Method on selection of random set

*Figure 2* required identification of matched, random individuals to assess the potential person-specific component of eyeAge predictions. For *Figure 2—figure supplement 3*, we created matched sets of visit pairs for each patient's first visit by identifying a randomly matching patient visit that was 0–2 years after a patient's first visit. To eliminate artifacts due to sampling differences between first and second visits, once we identified a patient's first visit to match, we constrained its set of potential randomly matched patient visits to only be from second visits. For the longitudinal analysis in 2B (right), individuals were split both by age and by time between visits (using 2 month buckets) and, again, randomly matched. For *Figure 2D*, the individuals were split evenly in 2-year buckets. Individuals within the same bucket had their left and right predictions compared to one another.

## Regression analyses in UK Biobank

Cox proportional hazards regression was performed using the lifelines package, https://github.com/CamDavidsonPilon/lifelines. Since retinal imaging was performed at the initial visit, individuals with events with an unknown date or date prior to the initial visit were excluded. All UK Biobank algorithmically defined outcomes with at least 4000 events were analyzed: asthma (field 42014), COPD (field 42016), dementia (field 42018), myocardial infarction (field 42000), all-cause Parkinsonism (field 42030), and stroke (field 42006). We note that because eyeAgeAccel is defined as eyeAge - alpha * age - beta for constants alpha and beta identified through regression of age on eyeAge, hazard ratios for eyeAge are identical to those in which eyeAgeAccel is used in the model instead.

Linear regression was performed on morbidity-related measurements taken at the same visit during which retinal imaging occurred, and was implemented using the statsmodels package with the model INT(outcome)~INT(age)+sex + INT(eyeAgeAccel), where INT(…) represents the rank-based inverse normal transformation. Individuals for which any of the outcome, age, or eyeAgeAccel variables were in the top 1% of outlier values were excluded. Measurements analyzed were: Overall health rating (field 2178), Long-standing illness (field 2188), Systolic blood pressure (field 4080), Diastolic blood pressure (field 4079), Pulse wave arterial stiffness index (field 21021), Health score (England) (field 26413), Fluid intelligence score (field 20016).

## GWAS

The eyeAgeAccel value defined above was used as the target for GWAS analysis. GWAS analysis was performed on the fundus-based phenotype as described previously (*Alipanahi et al., 2021*). Briefly, samples were restricted to individuals of European ancestry to avoid confounding effects due to population structure. European genetic ancestry was defined by computing the medioid of the 15-dimensional space of the top genetic principal components in individuals who self-identified as 'British' ancestry and defining all individuals within a distance of 40 from the medioid as 'European' (corresponding to the 99th percentile of distances of all individuals who self-identified as British or Irish). Samples were further restricted to those who also passed sample quality control measures computed by UK Biobank, that is those not flagged as outliers for heterozygosity or missingness, possessing a putative sex chromosome aneuploidy, or whose self-reported and genetically inferred sex were discordant.

BOLT-LMM v2.3.4 was used to examine associations between genotype and eyeAgeAcceleration in European individuals in the UK Biobank (n=45,444). All genotyped variants with minor allele frequency >0.001 were used to perform model fitting and heritability estimation. GWAS was performed in genotyped variants and imputed variants on autosomal chromosomes, with imputed variants filtered to exclude those with minor allele frequency (MAF) <0.001, imputation INFO score <0.8, or Hardy-Weinberg equilibrium (HWE) $P<1 \times 10^{-10}$ in Europeans. In total, 13,297,147 variants passed all quality control measures. Covariates included in the association study were chronological age, sex, genotyping array type, the top five principal components of genetic ancestry, and indicator variables for the six assessment centers used for the imaging.

Genome-wide suggestive ($p \leq 1 \times 10^{-6}$) lead SNPs, independent at $R^2 \leq 0.1$, were identified using the –clump command in PLINK version v1.90b4. The LD reference panel contained 10,000 unrelated UK Biobank subjects of European ancestry (as defined above). To identify distinct non-overlapping loci of association, all variants with $R^2 \geq 0.1$ with a lead SNP were grouped into a 'cluster' with that lead SNP, and subsequently clusters within 250 kilobases of each other were merged, with the lowest p-value lead SNP retained as the locus representative. Putative causal variants were identified using susieR version 0.9.0. At each locus containing at least 10 variants in LD, the susieR::susie_suff_stat function was used to estimate posterior inclusion probabilities for each variant in the locus, using the same LD reference panel as was used to generate loci and with a maximum of L=10 causal variants per locus and 200 iterations of coordinate ascent.

## Validation of Alk in fly
### Fly husbandry and phenotyping
For fly crosses, 15 virgin females were crossed with 3 males in bottles containing 1.55% live yeast, cornmeal, sugar, and agar (*Wilson et al., 2020*). Crosses were dumped 5 days following crossing, and female progeny were sorted into 4 replicate vials of 25 flies each, with food containing 200 μm RU486 to induce activation of the Gal-UAS system (*Nicholson et al., 2008*). Flies were maintained in 65% relative humidity at 25 °C in a 24 hr light/dark cycle throughout life. Two weeks post-induction, phototaxis was tested as previously described *Hodge et al., 2022* by placing flies in a clear, empty 30 cm-long vial horizontally in a dark room. Light was shined on one end and the number of flies in the last 10 cm closest to the light source after 1 min was scored for responsiveness to light signals. This was tested across each of the four vials per group in three biological replicates (total 100 flies per replicate). Strains used were 3x*elav*-GS (provided from the lab of Geetanjali Chawla) *Parkhitko et al., 2020* for RU486-dependent pan-neuronal Gal4, $w^{Dah}$ control strain, UAS-*Alk*$^{RNAi}$, and UAS-*Alk*$^{DN}$ (provided from the lab of Linda Partridge) (*Woodling et al., 2020*).

## Pathway analysis
All significant ($p<1.0 \times 10^{-6}$) GWAS candidates were used to assess pathway enrichment via Enrichr (*Xie et al., 2021*).

## Statistical analysis
For *Drosophila* phototaxis results, significance ($p<0.05$) was assessed using unpaired t-test. For *Figure 4C*, error bars represent SD across at least three biological replicates. Significant differences between experimental groups and controls are indicated by *. *, $p<0.05$. Statistical analyses were calculated with GraphPad Prism 4.

## Data and code availability
A subset of EyePACS data is freely available online (https://www.kaggle.com/competitions/diabetic-retinopathy-detection/data). To enquire about access to the full EyePACS dataset, researchers should contact Jorge Cuadros (jcuadros@eyepacs.com). The UK Biobank data are available for approved projects (application process detailed at https://www.ukbiobank.ac.uk/enable-your-research/apply-for-access) through the UK Biobank Access Management System (https://www.ukbiobank.ac.uk). We have

deposited the derived data fields and model predictions following UK Biobank policy, which will be available through the UK Biobank Access Management System. Full GWAS summary statistics are available in the Supplementary File. To develop the eyeAge model we used the TensorFlow deep learning framework, available at https://www.tensorflow.org. Code and detailed instructions for both model training and prediction of chronological age from fundus images is open-source and freely available as a minor modification (https://gist.github.com/cmclean/a7e01b916f07955b2693112dcd3edb60), (*Ahadi, 2023* copy archived at swh:1:rev:ba002c0a6edddd13814ecc9e07ec14249b2375f4) of our previously published repository for fundus model training (https://zenodo.org/record/7154413) (*Cosentino et al., 2021*).

## Acknowledgements

This research has been conducted with the UK Biobank resource application 17643. We thank Jorge Cuadros from EyePACS for data access and helpful conversations. KAW is supported by NIH T32AG000266-23. We thank the Bloomington *Drosophila* Stock Center for providing flies used in this study. This work is funded by grants awarded to PK from the Reta Haynes Foundation, American Federation of Aging Research, NIH grants R01 R01AG038688 and AG045835 and the Larry L Hillblom Foundation.

## Additional information

### Competing interests

Sara Ahadi, Ali Bashir: is not currently affiliated with Google Research, however work for this manuscript was conducted while affiliated with Google Research. The author has no other competing interests to declare. Boris Babenko, Cory Y McLean, Avinash Varadarajan: is affiliated with Google Health. The author has no other competing interests to declare. Drew Bryant, Orion Pritchard, Marc Berndl: is affiliated with Google Research. The author has no other competing interests to declare. Pankaj Kapahi: Reviewing editor, eLife. The other authors declare that no competing interests exist.

### Funding

| Funder | Grant reference number | Author |
| --- | --- | --- |
| NIH | T32AG000266-23 | Kenneth A Wilson |
| NIH | R01AG038688 | Pankaj Kapahi |
| NIH | AG045835 | Pankaj Kapahi |
| Larry L. Hillblom Foundation | | Pankaj Kapahi |

The funders had no role in study design, data collection and interpretation, or the decision to submit the work for publication.

### Author contributions

Sara Ahadi, Conceptualization, Data curation, Formal analysis, Supervision, Investigation, Visualization, Methodology, Writing – original draft, Project administration, Writing – review and editing; Kenneth A Wilson, Formal analysis, Validation, Visualization, Writing – original draft, Writing – review and editing, Investigation; Boris Babenko, Cory Y McLean, Data curation, Software, Formal analysis, Visualization, Methodology, Writing – original draft, Writing – review and editing; Drew Bryant, Orion Pritchard, Formal analysis; Ajay Kumar, Writing – original draft; Enrique M Carrera, Validation; Ricardo Lamy, Interpretation of results; Jay M Stewart, Interpretation of results; Avinash Varadarajan, Conceptualization, Resources, Data curation, Software, Formal analysis; Marc Berndl, Conceptualization, Formal

analysis, Supervision, Visualization, Methodology; Pankaj Kapahi, Conceptualization, Supervision, Funding acquisition, Validation, Methodology, Writing – original draft, Writing – review and editing; Ali Bashir, Conceptualization, Data curation, Formal analysis, Supervision, Investigation, Visualization, Methodology, Writing – original draft, Writing – review and editing

**Author ORCIDs**
Sara Ahadi http://orcid.org/0000-0002-7849-2135
Kenneth A Wilson http://orcid.org/0000-0003-3227-9977
Cory Y McLean http://orcid.org/0000-0001-9928-8216
Pankaj Kapahi http://orcid.org/0000-0002-5629-4947

**Ethics**
The UK Biobank study was reviewed and approved by the North West Multi-Centre Research Ethics Committee. For the EyePACS study, ethics review and IRB exemption was obtained using Quorum Review IRB (Seattle, WA).

**Decision letter and Author response**
Decision letter https://doi.org/10.7554/eLife.82364.sa1
Author response https://doi.org/10.7554/eLife.82364.sa2

---

## Additional files

**Supplementary files**
- Supplementary file 1. Hazard ratio results for men and women.
- Supplementary file 2. Hazard ratio results with adjustments.
- Supplementary file 3. Cox proportional hazards regression of Outcome on Age, Sex, and eyeAge. P-value and Hazard ratio are reported for eyeAge.
- Supplementary file 4. Linear regression of INT(Outcome) on INT(Age), Sex, INT(eyeAgeAccel).
- Supplementary file 5. Linear regression of visual acuity-related outcomes on age measurements.
- Supplementary file 6. Filtered gene association results.
- Supplementary file 7. Fine mapping gene association results.
- Supplementary file 8. List of genes associated with eyeAgeAccel and function.
- Supplementary file 9. Gene association results with annotated hits.
- MDAR checklist

**Data availability**
A subset of EyePACS data is freely available online (https://www.kaggle.com/competitions/diabetic-retinopathy-detection/data). To enquire about access to the full EyePACS dataset, researchers should contact Jorge Cuadros (jcuadros@eyepacs.com). Proposals and agreements are assessed internally at EyePACS and may be subject to ethics approvals. The UKB data are available for approved projects (application process detailed at https://www.ukbiobank.ac.uk/enable-your-research/apply-for-access) through the UK Biobank Access Management System (https://www.ukbiobank.ac.uk) . We have deposited the derived data fields and model predictions following UKB policy, which will be available through the UK Biobank Access Management System. Full GWAS summary statistics are available in the Supplementary File. To develop the eyeAge model we used the TensorFlow deep learning framework, available at https://www.tensorflow.org. Code and detailed instructions for both model training and prediction of chronological age from fundus images is open-source and freely available as a minor modification (https://gist.github.com/cmclean/a7e01b916f07955b2693112dcd3edb60, (copy archived at swh:1:rev:ba002c0a6edddd13814ecc9e07ec14249b2375f4)) of our previously published repository for fundus model training (https://zenodo.org/record/7154413).

The following previously published dataset was used:

| Author(s) | Year | Dataset title | Dataset URL | Database and Identifier |
|---|---|---|---|---|
| Cosentino J, Alipanahi B, Hormozdiari F, McLean CY | 2021 | Code for training fundus models | https://zenodo.org/record/7154413 | Zenodo, 10.5281/zenodo.7154413 |

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
