## [Editor Report]

This paper is an important contribution to the biological aging field using eye image data to create an aging clock of the retina in data from eyePACS with validation in the UK Biobank. The authors provide compelling evidence that the clock correlates with chronological and phenotypic age, predicting mortality independently of chronological age and showing longitudinal evidence. The work identifies novel genetic loci with a top site located in the ALKAL2 region, which is functionally validated in a *Drosophila* model.

---

## [Decision Letter]

**Decision letter after peer review:**

Thank you for submitting your article "Longitudinal fundus imaging and its genome-wide association analysis provide evidence for a human retinal aging clock" for consideration by *eLife*. Your article has been reviewed by 2 peer reviewers, one of whom is a member of our Board of Reviewing Editors, and the evaluation has been overseen by Carlos Isales as the Senior Editor.

Essential revisions:

1) Additional investigations on the age acceleration residuals are suggested, to differentiate between chronological and biological aging, which is needed.

2) Additional follow-up analyses in UKB as suggested.

*Reviewer #1 (Recommendations for the authors):*

The clock is trained on chronological age and age acceleration measures are compared between this clock and phenotypic age acceleration. However, another way would have been to train the clock on phenotypic age instead of chronological age. The second-generation epigenetic clocks are done in this way and those clocks perform better than the first-generation clocks. Since the authors use two sets of data, this approach could have been employed if starting with the UK Biobank where the phenoage clock is available. Could the authors please comment on this approach and discuss whether you think that your results could have been different?

Gender is a social construct while sex is the biological variable that is intended in this paper. Please change accordingly.

Please comment on sex differences. Are there differences in retinal aging between men and women? Is the eye clock performing better if trained separately in men and women? Whenever possible, please provide supplementary figures for sex-stratified analyses (Table 1, Figure 3D, etc.).

Figure 3D – please provide the full scale on the x-axis, and add the 95% Cis in numbers.

How is the pathway enrichment analysis done? I cannot find any information on that. Likewise, no statistical analysis section is included in the methods. A description of the Cox models is needed. Are the proportional hazards assumption met? How is age adjusted for in the model? What is the underlying time scale? Many more lifestyle factors etc. are available in UK Biobank that could additionally be adjusted for in the model.

The GWAS result, how does it compare to other GWAS of biological aging? There have been GWAS publications on phenotypic age, epigenetic clocks, telomere length, mitochondrial DNA abundance, etc.

For all figures: The figure legend should stand alone. Please provide additional information about the data used in the figure, sample sizes, methods used to perform the analyses, etc.

*Reviewer #2 (Recommendations for the authors):*

Below I have a few specific comments that, being addressed, would improve the article:

The background in para 1 requires a bit of revision for accuracy:

– In para 1, the article states that the PhenoAge is an algorithm derived from blood biomarkers based on chronological age. It is not. In fact, the PhenoAge includes chronological age as a component in addition to biomarker information. The biomarkers included were selected on the basis of their age-independent associations with mortality risk. The parenthetical text should be revised.

– Also in para 1, the authors refer to "an epigenetic clock". But the reference is to a review article that summarizes many of these clocks. I would suggest that they should simply refer to "clocks" plural.

– Finally the suggestion that these clocks require invasive cell or tissue extraction is a bit of an overstatement. What the existing measures of biological age do require is a blood draw + multiplex assay. The authors should just say that.

The longitudinal analysis reported in para 2 is very helpful. However, given the claims about the accuracy of the assessment, the authors should report how close the estimated change in age from the retinal measures was to the calendar time elapsed between measurements, not just the probability that the measurements were correctly ordered. (perhaps this is what they are reporting as MAE in this analysis. But if so, it should be clarified) Also, given the very large size of the EyePACS data, it would be feasible to exclude the longitudinal samples from the training entirely. This would make for a cleaner version of the longitudinal analysis, although I don't think this is necessary.

The UKB analysis comparing EyeAge and PhenoAge is also very helpful. However, it would be enhanced by including a set of analyses in which both EyeAge and PhenoAge were regressed on chronological age (separately) and residual values predicted. The correlation among these residuals will inform whether the biological aging information in the two measures is similar or different. In addition, it would be helpful to see effect sizes for associations of the residuals with mortality. (separate models for each measure with the inclusion of covariates for chronological age and sex would be ideal) These additional analyses can clarify (a) how similar the age-independent information in the two measures is, and (b) how they compare as mortality predictors.

[Editors' note: further revisions were suggested prior to acceptance, as described below.]

Thank you for resubmitting your work entitled "Longitudinal fundus imaging and its genome-wide association analysis provide evidence for a human retinal aging clock" for further consideration by *eLife*. Your revised article has been evaluated by Carlos Isales (Senior Editor) and a Reviewing Editor.

The manuscript has been improved but there are some remaining issues that need to be addressed, as outlined below by reviewer 2.

*Reviewer #1 (Recommendations for the authors):*

The authors have addressed the reviewer comments.

*Reviewer #2 (Recommendations for the authors):*

The authors have mostly addressed my comments. In particular, Figure 3 panels C and D are very helpful, as is the additional analysis of longitudinal change. I note below a few areas where further clarification could improve the manuscript. I reiterate my enthusiasm for the manuscript, which I think makes an important contribution to the literature on biological aging and, to the extent the authors measure can be readily implemented in retinal imaging data beyond the datasets reported in this manuscript, has the potential to deliver a new tool to aging research.

The only change I would argue is essential is revision of the statement in the abstract that EyeAge was 71% accurate in measuring aging in longitudinal data. As noted below, based on data reported by the authors, the accuracy of prediction of time elapsed between repeated measures is <3%. The claim in the abstract should be revised for clarity.

Further comments:

original reviewer comment: The longitudinal analysis reported in para 2 is very helpful. However, given the claims about the accuracy of the assessment, the authors should report how close the estimated change in age from the retinal measures was to the calendar time elapsed between measurements, not just the probability that the measurements were correctly ordered. (perhaps this is what they are reporting as MAE in this analysis. But if so, it should be clarified).

Author response: Correlation calculated between estimated change in chronological age and predicted age, while significant, is low (pearson Rho = 0.17, p-value = 1.4e-12). These differences may be due to the distinction of measuring chronological age vs. biological age as well as the inherent noise associated in the measurement. However, we agree that this information may be valuable to the reader and we have included this correlation number and p-value in the manuscript (changes underlined below).

"Using the EyePACS Tune dataset, we restricted ourselves to data from patients with exactly two visits (1,719 subjects) and examined the models' ability to order the two visits over multiple time scales. Note that no longitudinal information about patients was specifically used to train or tune the model to predict chronological age. While the observed and predicted age differences between the two visits had low correlation (pearson ⍴ = 0.17, p-value = 1.4e-12), …".

R1 reviewer comment: That correlation is quite low. However, to interpret it, we need a bit more information about how much variation there is in time elapsed between the two visits included in analysis. I wonder if a scatterplot as a supplemental figure would help clarify. Specifically, I am thinking that if there is little variation in the time elapsed, the correlation is likely to be very low. Why not instead report the distribution of the prediction error? In other words, what was the mean and SD of the difference between time elapsed and change in EyeAge?

In addition, the authors should revise the statement in the abstract:

"Longitudinal studies showed that the resulting models were able to predict individuals' aging in time-scales less than a year with 71% accuracy".

In fact, it appears that the accuracy of prediction is just slightly under 3% (i.e. 0.17^2). What the algorithm could do with 71% accuracy was correctly order two repeated observations in time. The abstract should be revised.

---

## [Author Response]

Reviewer #1 (Recommendations for the authors):The clock is trained on chronological age and age acceleration measures are compared between this clock and phenotypic age acceleration. However, another way would have been to train the clock on phenotypic age instead of chronological age. The second-generation epigenetic clocks are done in this way and those clocks perform better than the first-generation clocks. Since the authors use two sets of data, this approach could have been employed if starting with the UK Biobank where the phenoage clock is available. Could the authors please comment on this approach and discuss whether you think that your results could have been different?

The primary issues here are that the EyePACS not only lacks the measurements required for phenoAge but also lacks mortality and genomics data. It’s unclear what type of validation we could then perform on the EyePACs if we trained such a model. Similarly, splitting the UK Biobank data is problematic for two reasons: (1.) The dataset was already fairly limited for GWAS, further fragmentation would be inherently problematic, (2.) A key strength of the approach is cross-dataset generalization and within a single dataset we would not be able to make that claim.

Gender is a social construct while sex is the biological variable that is intended in this paper. Please change accordingly.

We thank the reviewer for bringing this to our attention. We used the following field: [https://biobank.ctsu.ox.ac.uk/crystal/field.cgi?id=31], which is self-reported sex. Throughout the updated manuscript, to make this clear, we have replaced references to this variable with "self-reported sex".

Please comment on sex differences. Are there differences in retinal aging between men and women? Is the eye clock performing better if trained separately in men and women? Whenever possible, please provide supplementary figures for sex-stratified analyses (Table 1, Figure 3D, etc.).

We appreciate the reviewer’s suggestion and have added the requested analysis for comparing hazard ratio stratified by sex. We have Supplementary File 1 which shows the Age and phenoAge adjustments for males, females, and combined.

Similarly, we have added the following text to the paper:

“Stratifying the hazard ratio analysis showed a slight increase in the hazard ratio for women compared to men (1.035 vs. 1.026), however the confidence intervals overlapped heavily (Supplementary File 1).”

Figure 3D – please provide the full scale on the x-axis, and add the 95% Cis in numbers.

We have added the full scale axis and the 95% CI’s to the text. See text below:

“The hazard ratio for eyeAgeAccel was was statistically significant when adjusting for (self-reported) sex (1.09, CI=[1.08, 1.10], p-value=1.6e-53), sex and age (1.04, CI=[1.02, 1.06], p-value=1.8e-4), and sex, age, and phenoAge (1.03, CI=[1.01, 1.05], p-value=2.8e-3) (Figure 3D).”

How is the pathway enrichment analysis done? I cannot find any information on that. Likewise, no statistical analysis section is included in the methods.

Pathway enrichment was performed via Enrichr online platform using all significant gene candidates from the GWAS. The literature for this platform has been cited in the text, and additional details regarding this as well as the statistical analyses have been added to the Methods.

A description of the Cox models is needed. Are the proportional hazards assumption met? How is age adjusted for in the model? What is the underlying time scale? Many more lifestyle factors etc. are available in UK Biobank that could additionally be adjusted for in the model.

Age has been adjusted in the Cox model by including it as a covariate, same as self-reported sex and phenotypic age. The unit of eyeAge is 1 year. We used the lifelines library to perform a Grambsch-Therneau test of proportional hazards assumption. At a p-value cutoff of 0.05, it did not reject the null hypothesis that the coefficients are not time-varying, using two time transforms (km and rank):

**Author response table 1. sa2table1:** 

		test_statistic	p	-log2(p)
chronological age	km	2.16	0.14	2.82
	rank	2.16	0.14	2.82
eyeAge	km	0.22	0.64	0.65
	rank	0.22	0.64	0.65
sex	km	2.47	0.12	3.11
	rank	2.47	0.12	3.11
phenoAge	km	2.63	0.11	3.25
	rank	2.63	0.10	3.25

The GWAS result, how does it compare to other GWAS of biological aging? There have been GWAS publications on phenotypic age, epigenetic clocks, telomere length, mitochondrial DNA abundance, etc.

We have compared our results to GWAS results on phenotypic age of UK Biobank participants and there’s no overlap between significant genes (Kuo et al. 2021). We suggest that is because eyeAge and phenotypic age each capture different aspects of biological aging and as our hazard ratio results indicate, they are independent of each other. We have added a citation to the Kuo et al. manuscript in the Discussion section.

For all figures: The figure legend should stand alone. Please provide additional information about the data used in the figure, sample sizes, methods used to perform the analyses, etc.

We have added the dataset used and analysis method to the figure legends as applicable.

Reviewer #2 (Recommendations for the authors):Below I have a few specific comments that, being addressed, would improve the article:The background in para 1 requires a bit of revision for accuracy:– In para 1, the article states that the PhenoAge is an algorithm derived from blood biomarkers based on chronological age. It is not. In fact, the PhenoAge includes chronological age as a component in addition to biomarker information. The biomarkers included were selected on the basis of their age-independent associations with mortality risk. The parenthetical text should be revised.

We thank the reviewer for the correction. We have updated the text to read:

“a combination of chronological age and 9 biomarkers predictive of mortality”.

– Also in para 1, the authors refer to "an epigenetic clock". But the reference is to a review article that summarizes many of these clocks. I would suggest that they should simply refer to "clocks" plural.

Thank you for the suggestion. As suggested, we have used “clocks” instead.

– Finally the suggestion that these clocks require invasive cell or tissue extraction is a bit of an overstatement. What the existing measures of biological age do require is a blood draw + multiplex assay. The authors should just say that.

Thanks for the comment. We have changed the sentence to “many require a blood draw and multiplex assay of many analytes”.

The longitudinal analysis reported in para 2 is very helpful. However, given the claims about the accuracy of the assessment, the authors should report how close the estimated change in age from the retinal measures was to the calendar time elapsed between measurements, not just the probability that the measurements were correctly ordered. (perhaps this is what they are reporting as MAE in this analysis. But if so, it should be clarified)

Correlation calculated between estimated change in chronological age and predicted age, while significant, is low (pearson Rho = 0.17, p-value = 1.4e-12). These differences may be due to the distinction of measuring chronological age vs. biological age as well as the inherent noise associated in the measurement. However, we agree that this information may be valuable to the reader and we have included this correlation number and p-value in the manuscript.

“Using the EyePACS Tune dataset, we restricted ourselves to data from patients with exactly two visits (1,719 subjects) and examined the models’ ability to order the two visits over multiple time scales. Note that no longitudinal information about patients was specifically used to train or tune the model to predict chronological age. While the observed and predicted age differences between the two visits had low correlation (pearson ⍴ = 0.17, p-value = 1.4e-12), …”

Also, given the very large size of the EyePACS data, it would be feasible to exclude the longitudinal samples from the training entirely. This would make for a cleaner version of the longitudinal analysis, although I don't think this is necessary.

We are unclear if this is the reviewer’s suggestion, but as a point of clarification, we did separate the train and tune sets by patient. Therefore, the longitudinal data should be clean in that there is no contamination of patients samples from the EyePACS train set.

The UKB analysis comparing EyeAge and PhenoAge is also very helpful. However, it would be enhanced by including a set of analyses in which both EyeAge and PhenoAge were regressed on chronological age (separately) and residual values predicted. The correlation among these residuals will inform whether the biological aging information in the two measures is similar or different.

We believe that Figure 3 addresses this point:

In addition, it would be helpful to see effect sizes for associations of the residuals with mortality. (separate models for each measure with the inclusion of covariates for chronological age and sex would be ideal) These additional analyses can clarify (a) how similar the age-independent information in the two measures is, and (b) how they compare as mortality predictors.

We acknowledge the importance of adjusting for chronological age, and the nuances associated with doing this in the context of the residuals which already have been adjusted for age.

We have specifically adjusted for chronological age in our hazard analyses. Performing hazard analyses with the residual (or “acceleration”) variables in places of eyeAge and/or phenoAge produces the exact same hazard ratios for these variables, but changes the hazard ratio for chronological age. We believe this is because eyeAgeAccel = (eyeAge – W*chronologicalAge – bias), where W and bias are the coefficients that are fit to produce the residual (and phenoAgeAccel is similarly defined). This introduces a clear collinearity between the variables, since chronological age is included multiple times with different weights. For this reason, we feel that it is more appropriate to perform mortality hazard analysis with eyeAge and phenoAge, rather than their acceleration counterparts.

[Editors' note: further revisions were suggested prior to acceptance, as described below.]

Reviewer #2 (Recommendations for the authors):The authors have mostly addressed my comments. In particular, Figure 3 panels C and D are very helpful, as is the additional analysis of longitudinal change. I note below a few areas where further clarification could improve the manuscript. I reiterate my enthusiasm for the manuscript, which I think makes an important contribution to the literature on biological aging and, to the extent the authors measure can be readily implemented in retinal imaging data beyond the datasets reported in this manuscript, has the potential to deliver a new tool to aging research.The only change I would argue is essential is revision of the statement in the abstract that EyeAge was 71% accurate in measuring aging in longitudinal data. As noted below, based on data reported by the authors, the accuracy of prediction of time elapsed between repeated measures is <3%. The claim in the abstract should be revised for clarity.

We thank the reviewer for their comment and have removed the following statement from the abstract to avoid any confusion:

“Longitudinal studies showed that the resulting models were able to correct individuals’ aging in time-scales less than a year with 71% accuracy.”

Further comments:Original reviewer comment: The longitudinal analysis reported in para 2 is very helpful. However, given the claims about the accuracy of the assessment, the authors should report how close the estimated change in age from the retinal measures was to the calendar time elapsed between measurements, not just the probability that the measurements were correctly ordered. (perhaps this is what they are reporting as MAE in this analysis. But if so, it should be clarified).Author response: Correlation calculated between estimated change in chronological age and predicted age, while significant, is low (pearson Rho = 0.17, p-value = 1.4e-12). These differences may be due to the distinction of measuring chronological age vs. biological age as well as the inherent noise associated in the measurement. However, we agree that this information may be valuable to the reader and we have included this correlation number and p-value in the manuscript (changes underlined below)."Using the EyePACS Tune dataset, we restricted ourselves to data from patients with exactly two visits (1,719 subjects) and examined the models' ability to order the two visits over multiple time scales. Note that no longitudinal information about patients was specifically used to train or tune the model to predict chronological age. While the observed and predicted age differences between the two visits had low correlation (pearson ⍴ = 0.17, p-value = 1.4e-12), …".R1 reviewer comment: That correlation is quite low. However, to interpret it, we need a bit more information about how much variation there is in time elapsed between the two visits included in analysis. I wonder if a scatterplot as a supplemental figure would help clarify. Specifically, I am thinking that if there is little variation in the time elapsed, the correlation is likely to be very low. Why not instead report the distribution of the prediction error? In other words, what was the mean and SD of the difference between time elapsed and change in EyeAge?

We’ve plotted the requested scatterplot showing the time elapsed (x-axis) vs. the difference between time elapsed and change in eyeAge (y-axis). We also calculated the distribution of the prediction error with a mean of 0.033 and standard deviation of 2.34. This plot was added to the manuscript as Figure 2—figure supplement 2.

With the text:

“While the observed and predicted age differences between the two visits (M = 0.033, SD = 2.34) had low correlation (pearson ⍴ = 0.17, p-value = 1.4e-12, Figure 2—figure supplement 2), Figure 2A shows that the model correctly ordered 71% of visits within a year with an MAE less than 2 years. In both metrics the fidelity decreased in older groups and with smaller age gaps.”

In addition, the authors should revise the statement in the abstract:"Longitudinal studies showed that the resulting models were able to predict individuals' aging in time-scales less than a year with 71% accuracy".In fact, it appears that the accuracy of prediction is just slightly under 3% (i.e. 0.17^2). What the algorithm could do with 71% accuracy was correctly order two repeated observations in time. The abstract should be revised.

We have removed the following statement from the abstract to avoid any confusion:

“Longitudinal studies showed that the resulting models were able to correct individuals’ aging in time-scales less than a year with 71% accuracy.”